# Impact of the Uncoupling Protein 1 on Cardiovascular Risk in Patients with Rheumatoid Arthritis

**DOI:** 10.3390/cells10051131

**Published:** 2021-05-07

**Authors:** Lovisa I. Lyngfelt, Malin C. Erlandsson, Mitra Nadali, Shahram Hedjazifar, Rille Pullerits, Karin M. Andersson, Petra Brembeck, Sofia Töyrä Silfverswärd, Ulf Smith, Maria I. Bokarewa

**Affiliations:** 1Department of Rheumatology and Inflammation Research, Institute of Medicine, University of Gothenburg, 405 30 Gothenburg, Sweden; malin.erlandsson@rheuma.gu.se (M.C.E.); mitra.nadali@vgregion.se (M.N.); rille.pullerits@rheuma.gu.se (R.P.); karin.andersson@rheuma.gu.se (K.M.A.); sofia.silfversward@rheuma.gu.se (S.T.S.); maria.bokarewa@rheuma.gu.se (M.I.B.); 2Rheumatology Clinic, Sahlgrenska University Hospital, 413 45 Gothenburg, Sweden; 3Lundberg Laboratory for Diabetes Research, Department of Molecular and Clinical Medicine, Institute of Medicine, University of Gothenburg, 405 30 Gothenburg, Sweden; shahram.hedjazifar@astrazeneca.com (S.H.); petra.brembeck@gu.se (P.B.); ulf.smith@gu.se (U.S.); 4Department of Clinical Immunology and Transfusion Medicine, Sahlgrenska University Hospital, 413 45 Gothenburg, Sweden

**Keywords:** cardiovascular risk, rheumatoid arthritis, adipose tissue, UCP-1, IL-6, tocilizumab

## Abstract

Adiposity is strongly associated with cardiovascular (CV) morbidity. Uncoupling protein 1 (UCP1) increases energy expenditure in adipocytes and may counteract adiposity. Our objective was to investigate a connection between *UCP1* expression and cardiovascular health in patients with rheumatoid arthritis (RA) in a longitudinal observational study. Transcription of *UCP1* was measured by qPCR in the subcutaneous adipose tissue of 125 female RA patients and analyzed with respect to clinical parameters and the estimated CV risk. Development of new CV events and diabetes mellitus was followed for five years. Transcription of *UCP1* was identified in 89 (71%) patients. *UCP1* positive patients had often active RA disease (*p* = 0.017), high serum levels of IL6 (*p* = 0.0025) and were frequently overweight (*p* = 0.015). IL-6^hi^BMI^hi^ patients and patients treated with IL6 receptor inhibitor tocilizumab had significantly higher levels of *UCP1* compared to other RA patients (*p* < 0.0001, *p* = 0.032, respectively). Both UCP1^hi^ groups displayed unfavorable metabolic profiles with high plasma glucose levels and high triglyceride-to-HDL ratios, which indicated insulin resistance. Prospective follow-up revealed no significant difference in the incidence of new CV and metabolic events in the UCP1^hi^ groups and remaining RA patients. The study shows that high transcription of *UCP1* in adipose tissue is related to IL6-driven processes and reflects primarily metabolic CV risk in female RA patients.

## 1. Introduction

Traditionally viewed as a passive reservoir for energy storage, adipose tissue (AT) is known to be a metabolically active endocrine organ performing critical biochemical reactions to utilize lipids, carbohydrates, and steroids. AT has emerged as an immunologically active organ, producing numerous cytokines and other signal molecules promoting inflammation. Malfunction of AT results in its quantitative expansion and makes a major contribution to the development of type 2 diabetes (T2D) and cardiovascular disease (CVD) [1]. One specific and distinctly separate functional quality of AT depends on efficacy of the thermogenic subgroup of adipocytes (also called brown adipocytes). Thermogenic AT is associated with a favorable increase in energy expenditure followed by changes in lipoprotein and carbohydrate metabolism. Activation of the thermogenic AT could quickly provide sufficient amount of basic caloric needs, which could be metabolically protective [2]. The functional properties of thermogenic AT are tightly related to the expression of the uncoupling protein 1 (UCP1), the active and heat-producing component of the adipocyte [3]. UCP1 disconnects (uncouples) the oxidative phosphorylation from the inner membrane in mitochondria generating a proton leakage in the mitochondria that consequently causes the energy of the proton gradient to degrade into thermal energy. These heat generating abilities of AT are normally induced in response to cold exposure but can also be activated by exercise and caloric restrictions. Since activation of thermogenic AT boosts metabolism and potentially protects against the obesity-related pathology, it has been attributed therapeutic properties operating through lowering serum levels of lipoproteins and improving insulin sensitivity [4,5,6]. However, *UCP1* expression has also been discussed as an unwanted metabolic expender. In cachexia, a frequent clinical sign in chronic inflammation, cancer, and anorexia nervosa, UCP1 is emerging as a mediator of unwanted wasting of muscle tissue [7,8]. It has been hypothesized that the inflammatory state in of itself could induce metabolic changes. Recent experimental studies on cancer tissue and on the healing tissue beneath burns showed that inflammation, provoked by high level of IL-6, increase expression of *UCP1* and predispose to catabolic changes in AT [7,9].

Rheumatoid arthritis (RA) is a relatively common joint disease characterized by systemic inflammation and an unexplained CV burden. Metabolic activity of AT has been implied to fuel the molecular pathology of RA [10,11,12]. This attracts particular attention to the quantity and quality of AT in RA patients. Several epidemiologic studies indicated AT and obesity to be risk factors for the onset of RA [13,14,15], while others failed to confirm this [16]. Among the patients with established RA, obesity is associated with higher disease severity including higher clinical and serological signs of inflammation [17]. In addition to the increased RA disease activity, obesity has been postulated to support resistance to anti-rheumatic drugs and lower the probability of remission [18]. These observations are consistent with the endocrine activity of AT, which grows in direct proportion to obesity. AT produces IL-6, TNF-α and adipokines among other numerous signal molecules [19]. In spite of the increased systemic inflammation, active AT is associated with a decreased joint and skeletal damage and is supposed to be protective for joints [20,21]. Additionally, overweight is associated with lower mortality of RA patients. The RA patients with low body mass have higher morbidity and mortality risk compared to those with normal weight [22]. These two sides of AT, protecting against metabolic diseases and aggravating inflammatory conditions, are especially interesting in RA, due to the fact that rheumatoid cachexia is characterized by a waste of the muscle tissue in the presence of stable or increased AT, remains among frequent comorbidities of RA [23]. A connection between inflammation and metabolic properties of AT remains largely unexplored. We hypothesized that expression of *UCP1* in AT represents an operative mechanism favoring metabolic health in female RA patients.

The primary objective of this study was to explore clinical and metabolic associates of *UCP1* expression in AT of female patients with RA. Secondary objectives were (a) to describe effect of anti-rheumatic treatment on *UCP1* expression; (b) to define if *UCP1* expression improves estimation of CV risk in RA patients; (c) and finally, to demonstrate if *UCP1* expression contributes to reduction of actual CV events.

## 2. Materials and Methods

### 2.1. Patients

In total, 129 female patients with established RA diagnosis who fulfilled the American Rheumatism Association 1987 revised criteria [24] were included in the study. One hundred six patients were included between November 2011 and September 2013, and the study was later completed with another 23 patients, mainly treated with tocilizumab, between September and December 2018. Patients were recruited at the Rheumatology Clinic of the Sahlgrenska University Hospital in Gothenburg and the Northern Älvsborg County Hospital in Uddevalla in Sweden from the list of RA patients treated with methotrexate (MTX). All but 12 patients, who had discontinued MTX for longer than 30 days before blood sampling, were treated with MTX. At enrolment, 70 patients were treated with biologics including infliximab (*n* = 35), rituximab (*n* = 8), etanercept (*n* = 13), adalimumab (*n* = 3), golimumab (*n* = 5), tocilizumab (*n* = 18), abatacept (*n* = 1). Additional 25 patients combined MTX with other disease modifying antirheumatic drugs (DMARD). Low dose (median 5.0 mg/day) of oral corticosteroids was used by 25 patients. All patients completed a structured questionnaire regarding their smoking habits, medication, and concomitant diseases. At inclusion, all patients underwent clinical examination performed by experienced rheumatologists. The following clinical and laboratory data were recorded: age, sex, body mass index, disease duration, and the presence of rheumatoid factor and anti-CCP antibody positivity. Disease activity of RA was calculated based on the evaluation of swelling and tenderness in 28 joints (DAS28) and the erythrocyte sedimentation rate (ESR). Exclusion criteria were male gender, rheumatologic diagnosis other than RA, and juvenile form of arthritis. 

The study was approved by the Swedish Ethical Review Authority (diary no. 659–11) and performed in accordance with the Declaration of Helsinki. All patients gave informed written consent prior to participation. The trial is registered at ClinicalTrials.gov with ID NCT03449589.

### 2.2. Cardiovascular Risk Assessment

#### 2.2.1. Pocock’s Risk Score

The risk of dying within 5 years in CV event was calculated with the Pocock’s Risk Score using a digital calculator [25]. The values of age, sex, former or current cigarette smoker, systolic blood pressure (BP), total cholesterol (TC), creatinine, and height were collected. In the calculations, information about diabetes mellitus, left ventricular hypertrophy, and history of myocardial infarction or stroke was also taken into account. Systolic BP was unknown for 15 patients and calculated as 120, the ages of 22 patients was below 35 years, but were calculated as 35, and the TC of 2 patients was below 3 mmol/L but was calculated as 3. The Pocock’s risk score above 40 predicts a high risk of dying due to CV event within 5 years.

#### 2.2.2. Framingham Risk Score

The risk of having a CV event within 10 years, FRS scores were calculated based on the six coronary risk factors including age, gender, TC, HDL-cholesterol, systolic blood pressure, and smoking habits [26]. Absolute CVD risk percentage over 10 years was classified as low risk (<10%), intermediate risk (10–20%), and high risk (>20%).

### 2.3. Prospective Cardiovascular Follow-Up

The mapping of actual CV events during the period of five years after the initial assessment was performed. The initial patient control was done from November 2011 to May 2013 and the follow up between March and May 2018. This follow up was done through a structured telephone interview or by a questionnaire sent through conventional post (89%). The remaining 11% were only checked against the Swedish National Patient Register. In all cases of CV events, the obtained information was confirmed through the Swedish National Patient Register. The information that was gathered was of CV risk factors that might have been added since the initial study and of actual cardiac events. Specifically, the patients were asked about drugs against hypertension, hypercholesterolemia, T1D and T2D, surgeries or history of thromboembolic events. Arrhythmias, such as atrial fibrillation and other cardiac events were also noted.

### 2.4. Collection and Preparation of Blood and Fat Tissue Samples

The serum, plasma and AT samples were obtained at the study visit after an overnight fast. Blood specimens were drawn from the cubital vein directly into vacuum tubes, mixed thoroughly on BioMixer (Sarstedt, Nurnbrecht, Germany), centrifuged at 2000× *g* for 10 min, and aliquoted. The aliquots were stored at −70 °C until use.

Abdominal wall AT biopsies were obtained by needle aspiration of subcutaneous fat tissue in the periumbilical area, as previously described [27,28]. In brief, a region 5 cm lateral from the umbilicus (either to the left or right side of the abdomen) was sterilized. A hypodermic needle measuring 1.2 × 40 mm (18G) was then adapted to a 20 mL syringe and the piston was compressed. Approximately one-third of the length of the needle was inserted into the subcutaneous fat, and the needle piston was released maximally, thereby creating a vacuum. Tissue resistance was created by the physician gripping the abdominal wall with one hand while the other hand moved the needle throughout the tissue. Once the sample was aspirated into the syringe, the needle was withdrawn and the piston was removed. Fat tissue samples were preserved in AllProtect (Qiagen, Valencia, CA, USA) and stored in −80 °C.

### 2.5. Preparation and Stimulation of Primary Adipocyte Cultures

Primary human adipocytes were prepared as described [29]. In short, aspirated adipose tissue was carefully washed with warm Hank’s medium 199 (pH 7.4, 22350029, Life Technologies, Carlsbad, CA, USA) supplemented with 5% BSA (Hyclone, Lund, Sweden) and 0.31 mg/mL of collagenase A (Roche, Basel, Switzerland) and incubated for 50 min in a shaking 37 °C water bath to allow the stromal fraction to digest. AT was thereafter filtered through the sterile 250 µm nylon filter to obtain monocellular culture and washed 4 times with pre-warmed Hank’s medium. The floating adipocyte cell layer was diluted to 1:3 in Hank’s medium supplemented with 4% BSA and metformin. The adipocyte cultures were activated for 5 h at 37 °C with recombinant IL6 (50 ng/mL, PeproTech, London, UK) in the presence of monoclonal humanized antibodies against IL6 receptor (1 mg/mL, tocilizumab (RoActemra^®^), Roche, Basel, Switzerland). The adipocytes were isolated by spinning the suspension through Dinonyl Phthalate oil (Alfa Aesar Chemicals, Haverhill, MA, USA). The adipocytes pellet was diluted in 350 µL RLT buffer (Qiagen) for future RNA extraction.

### 2.6. Preparation of mRNA

RNA from fat tissue, stored in Allprotect Tissue Reagent (Qiagen), was prepared with RNeasy Plus Universal Kit (Qiagen). The concentration and quality of the RNA were evaluated with a NanoDrop spectrophotometer (Thermo Scientific, Waltham, MA, USA) and Experion (Bio-Rad laboratories Inc., Hercules, CA, USA). A total of 400 ng RNA was used for cDNA synthesis using a High Capacity cDNA Reverse Transcription Kit (Applied Biosystems, Foster City, CA, USA).

### 2.7. Gene Expression Analysis

Real-time polymerized chain reaction (RT-PCR) was performed on a ViiA™7 RT-PCR (Applied Biosystems, Foster City, CA, USA) using the customer-designed multi-analyte array with primers for the *STAT3*, *RETN*, *ACTB* and *POLR2A* genes, and SYBR Green qPCR Mastermix (SA Biosciences, Qiagen). Analysis of *UCP1*, *LEP* and *ABCA1* genes was performed separately and normalized to *ACTB* gene (TATAA Biocentre, Gothenburg, Sweden). The primer pairs can be found in Appendix A. Melting curves for each PCR were performed between 60 °C and 95 °C to ensure specificity of the amplified product. Transcription of *UCP1* above 33 cycles threshold was considered undetectable and for comparative purposes was put as dCT 20.

### 2.8. Serological Parameters

The serum levels of resistin, adiponectin, leptin, IL4, IL9 and IL10 were determined using specific sandwich ELISA kits (R&D Systems, Minneapolis, MN, USA). The minimum detectable levels were 10 pg/mL for resistin (DY1359), 62 pg/mL for adiponectin (DY1065), 31 pg/mL for leptin (DY398), 0.2 pg/mL for IL4 (DY204), 0.0004 pg/mL for IL9 (DY209) and 0.0003 pg/mL for IL10 (DY217B). Serum levels of visfatin were measured by an ELISA kit (AG-45A-0006TP-KI01) purchased from Adipogene Inc (Incheon, South-Korea). The minimum detectable visfatin levels were 125 pg/mL. The sandwich ELISA for IL6 and IFNγ (Sanquin, Amsterdam, the Netherlands) had minimum detectable level of 0.075 pg/mL for IL6 (M9316) and 0.001 pg/mL for IFNγ (M1933). All assays were performed according to the manufacturers’ instructions. ELISA plates were read with a Spectramax 340 from Molecular Devices (Sunnyvale, CA, USA).

Plasma glucose levels were measured using FreeStyle Lite (Abbott Diabetes Care Ltd., Oxon, UK). Sandwich ELISAs were used to measure insulin (DY8056, R&D Systems, Minneapolis, MN, USA). Homeostatic model assessment for insulin resistance (HOMA-IR) was calculated. Levels of lipoproteins TC, low density lipoprotein (LDL), high density lipoprotein (HDL), and triglycerides (TG) were measured in serum after the overnight fasting period by photometry on Cobas 8000 (Roche Diagnostics, Switzerland) line at the laboratory of Clinical Chemistry, Sahlgrenska University Hospital, Gothenburg as described [27].

### 2.9. Statistical Analysis

Body mass index (BMI) above 25 kg/m^2^ and serum levels of IL6 above 1 pg/mL were defined as high. Absolute values are presented as the median (interquartile range) or number (%). Differences between unpaired groups were calculated by the Mann–Whitney U-test. Comparison between three groups was done using non-parametric Kruskal–Wallis statistics with Dunn’s multiple comparison post hoc analysis. Bivariate correlation between variables was examined by the Spearman’s correlation test. Differences between correlations were studied with Fisher’s r-to-z test. Relative risk was calculated as the odds ratio using 2 × 2 table analysis and mixed chi-square tests at OpenEpi software. Odds ratios are presented as OR (CI 95%). All further statistical evaluation of data was performed using GraphPad Prism version 8 (Chicago, IL, USA). All tests were two tailed and *p* < 0.05 was considered statistically significant.

## 3. Results

### 3.1. UCP1 Transcription in Adipose Tissue Associates with High Serum Levels of IL6 and Overweight

Transcription of *UCP1* in AT was measured in 111 female RA patients and revealed measurable *UCP1* mRNA in 80% (UCP1+, *n* = 89), while the remaining samples had no detectable transcription of *UCP1* (UCP1−, *n* = 22). Clinical characteristics of these patient groups are shown in Table 1. Demographically, the groups of UCP1+ and UCP1− patients comprised middle-aged females, they were similar with respect to age, disease duration and hormonal medication with oral corticosteroids, estradiol and levothyroxine that could have affected *UCP1* transcription.

We observed that the UCP1+ group had higher disease activity expressed by DAS28 (Figure 1A) and included all 17 patients with high disease activity (DAS28 > 5.1) (*p* = 0.033). UCP1+ group had also significantly higher systemic inflammation measured by ESR (Figure 1B) and serum levels of IL-6 (Figure 1C), while serum levels of other pro-inflammatory cytokines IL-1β (Figure 1D), IL-9 (Figure 1E) and IFNγ (Figure 1F) were not different between the groups. UCP1+ patients had higher body weight (Figure 1G) and a significantly increased frequency of overweight (BMI >25 kg/m^2^, 50/89 vs. 6/22, *p* = 0.017, OR 3.38) (Figure 1H).

Since both IL6 and BMI were associated with high *UCP1* transcription, we combined these two parameters (Figure 1I). Comparison of *UCP1* transcription in the formed groups showed that the frequency of UCP1+ and its transcription levels were significantly higher in IL6^hi^BMI^hi^ group compared to all other patients (Figure 1J) (39/42 vs. 50/69, OR 4.88 (1.45–22.01), *p* = 0.0079), and particularly in comparison to the IL6^lo^BMI^lo^ group (15/26, OR 9.19 (2.36–45.96), *p* = 0.00086). By grouping high *UCP1* transcription with its closest associated parameters BMI and IL6, we identified 39 patients who combined those parameters (HiHi UCP1+). Comparing HiHi UCP1+ patients to all other patients of the cohort, we observed significantly higher serum levels of AT products leptin (Figure 2A), resistin (Figure 2B) and visfatin (Figure 2C), all with known connection to IL-6. The AT of HiHi UCP1+ patients had high transcription of leptin (Figure 2D), and the transcription factor *STAT3* (Figure 2E), which mediates intracellular signaling down-stream of IL6 receptor. There was no difference for resistin (Figure 2F). Taken together, these results indicate that IL6 could have a direct stimulatory effect on UCP1 transcription in adipocytes and that transcription of UCP1 represented increased proportion of active thermogenic adipocytes in HiHi UCP1+ RA patients.

We further analyzed a relationship between active thermogenesis by UCP1 and reverse cholesterol transport driven by ABCA1. We found that expression of *ABCA1* was decreased in HiHi UCP1+ patients compared to all other patients (Figure 2G), while we found no significant correlation between transcription of *UCP1* and *ABCA1* in AT (Spearman’s r 0.03). *UCP1* correlated to leptin. *ABCA1* correlated significantly only to leptin and visfatin, opposing correlation between *UCP1* and leptin (r-to-z, *p* = 0.0001. Figure 2H).

Intrigued by the strong connection between *UCP1* transcription and IL6, we performed an in vitro experiment to study a causal connection between these parameters. Cultures of primary adipocytes were prepared from the abdominal AT of 7 healthy females (mean age 60 y, BMI 27.2) and stimulated with recombinant IL6 for 5 h. All 7 cultures presented measurable transcription of *UCP1* (dCT range 9.97–14.40). However, we could neither observe any change in *UCP1* nor *ABCA1* transcription after this short IL6 stimulation. Introduction of IL6R inhibitor tocilizumab to the IL6 stimulated cultures had no significant effect on *UCP1* and *ABCA1* transcription in those cells (Figure 2I,J).

### 3.2. Modulation of IL-6 Receptor Signaling Changes UCP1 Transcription

To pursue the association between the *UCP1* transcription and high IL-6 levels, we collected AT of 18 additional female RA patients who received regular treatment with tocilizumab, a monoclonal antibody against IL6 receptor. Demographically, these patients were similar with the other groups in age but had somewhat longer disease duration (Table 1). We compared *UCP1* transcription of the tocilizumab-treated patients with 84 patients treated with MTX, conventional (c)DMARDs (sulfasalazine, hydroxychloroquine, leflunomide, cyclosporine) and TNF inhibitors (TNFi). We found that *UCP1* mRNA was significantly higher in the tocilizumab-treated patients compared to the other groups (*p* = 0.032. Figure 3A). The prevalence of UCP1+ cases comprised 86% in the tocilizumab-treated patients, which was comparable to the patients treated with methotrexate mono therapy (*n* = 45, 73%), combination of MTX with TNFi (*n* = 28, 79%), and with cDMARDs (*n* = 17, 100%) (Figure 3A). Comparing UCP1+ patients, the tocilizumab-treated group had strikingly higher levels of serum IL6 compared to the patients treated with MTX, TNFi and cDMARD (*p* < 0.0001) (Figure 3B). Since transcription of *UCP1* was associated with BMI, we analyzed this in the tocilizumab-treated patients, but found no significant difference from other groups (Figure 3C).

Thus, we identified two groups of RA patients with high transcription of UCP1 in AT. These were patients with overweight, and patients treated with tocilizumab. High serum IL6 was the common feature for both of these groups.

### 3.3. Unfavourable Metabolic Profile in RA Patients with High Transcription of UCP1

In the next step, we assessed clinical significance of high *UCP1* transcription in AT. To do so, we analyzed carbohydrate and lipid metabolic profile of the groups with high transcription of *UCP1*. The HiHi UCP1+ group comprised patients with different anti-rheumatic treatment (*n* = 39) and tocilizumab-treated UCP1+ patients (*n* = 16) were compared to the UCP1- patients treated with MTX, cDMARDs and TNFi (*n* = 69).

The inflammatory activity in tocilizumab-treated patients was limited to high IL6 (Figure 3D), while ESR (Figure 3E) and DAS28 (Figure 3F) were significantly lower. Serum IFNγ (Figure 3G) and IL9 (Figure 3H) were comparable to other groups. Serum levels of anti-inflammatory cytokines IL10 (Figure 3I) and IL4 (Figure 3J) were significantly lower in tocilizumab-treated patients compared to HiHi UCP1+ patients.

We found that HiHi UCP1+ patients had significantly higher plasma glucose compared to all other patients (Figure 4A). Each group contained patients treated for diabetes mellitus (DM). The HiHi UCP1+ group included 3 patients with T2D and 1 patient with T1DM. The tocilizumab-treated UCP1+ group had 1 patient with T2D and 1 patient with T1DM, while the other patient group included 1 T2D and 1 T1DM patients. However, the difference in plasma glucose levels remained significant after the exclusion of those patients (*p* = 0.0002). Both UCP+ groups had higher insulin resistance expressed in HOMA-IR compared to other patients (Figure 4A,B). Cholesterol transporter ABCA1 were also significantly higher in HiHi UCP1 group (Figure 4C). Analysis of the lipid profile in serum taken after an overnight fast showed that the tocilizumab-treated UCP1+ patients had significantly higher TC and LDL levels compared to the HiHi UCP1+ patients (Figure 4D,E), whereas the levels of HDL were similar between groups (Figure 4F). Additionally, both UCP1+ groups had higher levels of triglycerides (Figure 4G) and presented an increased TG/HDL ratio compared to other patients (Figure 4H). High TG/HDL denotes an insufficient reverse cholesterol transport [30], which in our study was supported by low transcription of ABCA1 in HiHi UCP1+ patients (Figure 4D).

To analyze the effect of IL6 receptor inhibition on function of AT, we correlated UCP1 and ABCA1 transcription to the metabolic profile of RA patients. We observed that UCP1 and ABCA1 exhibited different, often opposing correlations in RA patients (Figure 4I). In all RA patients, there was a significant difference between UCP1 and ABCA1 for BMI, HOMA-IR, triglycerides, and TG/HDL. This analysis allowed disconnecting metabolic (UCP1-related) and lipid (ABCA1-related) CV risk parameters. Interestingly, tocilizumab-treated patients acquired strong negative correlation between the transcription of UCP1 and ABCA1 (r = −0.63, *p* = 0.024), which did not exist in RA patients treated with other anti-rheumatic drugs (r = 0.03).

### 3.4. High UCP1 Transcription Provided No Protection against Cardiovascular Events

To study if *UCP1* transcription contributes to CV morbidity, we estimated the risk to develop or die of CV disease by using the Pocock’s and Framingham risk models [31,32]. As a consequence of adverse metabolic profile and serum lipid levels, both UCP1+ groups had somewhat higher CV risk compared to other patients. In the Pocock’s model, this difference did reach significance (Kruskal–Wallis post hoc, *p* = 0.044) (Figure 4J). In the Framingham lipids model, the risk score was significantly higher in HiHi UCP1+ compared to the other patients (*p* = 0.011), while the tocilizumab-treated UCP1+ patients were similar to the others (*p* = 0.90) (Figure 4K).

To translate the estimated CV risk into actual occurrence of CV and metabolic events, we performed a 5-year follow-up of CV health in these patients. The prevalence of previous CV events was similar between the UCP1+ groups and the other patients (Figure 4L). In the prospective follow up of the remaining event-free patients, we identified 4 new CV events in HiHi UCP1+ patients. One of those was diagnosed with T2D (13.8%). Two new CV events were identified among other patients (3.4%). This translated into a 7.4% higher overall risk for development of CV and metabolic events in the HiHi UCP1+ patients (OR 4.52 (0.84–35.32), *p* = 0.081). Only 5 UCP1+ tocilizumab-treated patients completed the 5-year follow up. In that group, one CV event was registered. Four of the remaining tocilizumab-treated patients discontinued tocilizumab treatment within the first year.

These prospectively obtained results supported the CV risk estimations obtained by the Framingham model despite that the prevalence of CV and metabolic events failed to achieve statistically significant difference. It also let us to conclude that high *UCP1* transcription in AT provided no direct CV advantage for those RA patients.

## 4. Discussion

Estimation of CV risk in patients with RA often meets difficulties. Traditional prediction models provide only partial explanation to the well-documented premature CV mortality in RA patients [32]. Persistent disease activity and excess of inflammation initiated by cytokines has been postulated central for development of vascular pathology in RA [33]. In RA patients, thermogenic protein UCP1 is located on the crossroad of traditional metabolic CV risk and inflammation. In this study, we explored if *UCP1* transcription in AT could provide additional clues in assessment of CV risk in female RA patients. We observed an intimate relationship between the transcription of *UCP1* in AT and inflammation. Importantly, *UCP1* production was solely associated with inflammation driven by IL6, one of the major players in RA pathogenesis, while other pro-inflammatory cytokines IL1β, IFNγ and IL9 showed no such an association. The bond between IL6 and UCP1 was strengthened by the findings that high IL6 levels boosted by the inhibition of IL6 receptor were efficient in up-regulating *UCP1* production in AT of tocilizumab-treated patients.

Our findings demonstrated an association between *UCP1* transcription and inflammation, which was conveyed through high serum levels of IL6 and clinically active RA disease. Further analysis revealed that serum levels of IL6 were functionally relevant for AT and were translated in the activation of STAT3, the major intracellular mediator of IL6 effects and an important inducer of thermogenic AT, and the production of signaling molecules leptin, resistin and visfatin [34,35]. Studies on the association between UCP1 and inflammation are scarce. The relationship between inflammation and UCP1 has been reported in recent studies on cancer and burns [7,8,36,37]. In concordance with our results, high UCP1 production triggered by IL6 was proposed to be the leading mechanism for lipolysis and cachexia in cancer patients [7]. Using IL6-deficient mice, UCP1 was experimentally confirmed to be a reason for catabolic consumption of AT, since IL6-deficient mice were unable to up-regulate UCP1 and were protected from the weight loss [7,37,38]. A different study described UCP1 expression as a reason for increased resting energy expenditure in severely obese individuals, a condition characterized by chronic low-grade inflammation driven by IL6 [39]. The impact of UCP1 production on AT catabolism was less pronounced in these study patients. In contrast, UCP1 producers had higher BMI and body fat content compared to the remaining patients.

The expectations that UCP1 could play a protective role in CV health were based on the collected experimental evidence and epidemiological reports, which demonstrated an inverse association between UCP1 transcription and insulin resistance, dyslipidemia and obesity [40]. In fact, transcription of UCP1 was recognized in RA patients with unfavorable metabolic profile including overweight, insulin resistance and poor serum lipid profile suggesting that thermogenic conversion of energy was not enough to restore metabolic health in those patients. We observed that correlations between UCP1 production and metabolic CV risk factor were different if patients were treated with IL6 receptor inhibitor tocilizumab. In the tocilizumab-treated patients, UCP1 production was associated with high total and LDL cholesterol and high triglyceride levels. In contrast, cholesterol levels were low in UCP1 producing patients not treated with tocilizumab, while plasma glucose levels were high. High plasma glucose was further translated into insulin resistance expressed by high HOMA-IR and triglycerides/HDL ratio. This sequence of events induced by chronic exposure to high IL6 levels diverts from the expected associations reported for UCP1. Indeed, UCP1 production facilitated glucose intake by AT, which led to higher resting energy expenditure measured in the general population and in T2D patients [41]. Induction of UCP1 production by cryo-stimulation in healthy volunteers and in diabetic mice increased intracellular consumption of cholesterol and reduced its serum levels [2]. Ectopic UPC1 expression was sufficient to improve glucose metabolism in AT [6]. This exemplifies a disconnect between UCP1 production and mitochondrial energy expenditure under the condition of IL6-driven inflammation. In this study we show that IL6-driven UCP1 production is mediated by STAT3 in AT, despite the fact that direct stimulation of human primary adipocyte cultures with IL6 caused neither a change in UCP1 nor ABCA1 production. This strengthens our recent observations of independent association between CV risk and the expression of *STAT3* and leptin in AT [28] and emphasizes the importance of chronic IL6 exposure and neighboring macrophages and extracellular stroma for UCP1 production [9,42,43].

To find an explanation to these unexpected findings, we measured AT production of *ABCA1*, the key lipid transporter across cell membrane responsible for reverse cholesterol transport. We found *ABCA1* to be low in RA patients with high *UCP1* unless they were treated with tocilizumab. Dyslipidemia is a frequent complication of tocilizumab treatment in RA [44]. Interestingly, high production of *UCP1* in tocilizumab-treated patients neither counteracted the *ABCA1* increase nor prevented extracellular lipid efflux in those patients. Thus, we concluded that UCP1 and ABCA1 represented two independent mechanisms regulating and keeping in balance carbohydrate and lipid metabolism in AT and account for distinct CV risk parameters.

Hyperlipidemia is traditionally viewed as the major CV risk factor. This fits poorly with RA patients, who combine high CV morbidity with inflammatory burden and low total and LDL cholesterol levels, the phenomenon known as “lipid paradox” [45]. This study is not an exception. Highest levels of lipids are registered in the group of tocilizumab-treated patients, who have no signs of inflammation or disease activity. These patients have also low estimated CV risk, which is in accordance with recent report of the absence of excess in CV morbidity in tocilizumab treated RA patients [46]. In contrast, UCP1+ patients not treated with tocilizumab had higher CV risk. Importantly, the latter patient group had higher rate of actual CV events at a 5-year follow-up. The parameters associated with insulin resistance, including BMI, HOMA-IR index and TG/HDL ratio, were the major contributors to CV morbidity in those RA patients. This is in concordance with our previous report, where high IL6 and insulin predicted the development of early CV events in RA females [12,27,28]. However, the expression of UCP1 could be viewed as a protective mechanism developed to counteract insulin resistance and prevent development of classical metabolic syndrome in RA patients. The state of insulin resistance has been attracting lack of attention in RA despite its high prevalence and potential impact in the basic molecular mechanisms of joint inflammation [47,48]. This study investigates the molecular mechanisms behind metabolic and lipid CV risk factors and shows their direvsity. UCP1 reflects the metabolic side.

Notably, we did not observe any measurable protective effect of UCP1 production against coming CV events. The limited size of the studied RA cohort and the relatively good CV health, which resulted in only a few new events during the observation period, which could have contributed to these results. In our study, we rely on measures of the transcription of *UCP1* rather than its protein levels. Transcription of *UCP1* has previously been described to reflect UCP1 levels but the mRNA is not in itself thermogenic [49]. We performed no measurements of resting energy expenditure to visualize the actual thermogenic and metabolic potential of AT in the studied RA patients. This decision was at least, in part, justified by an appreciated lack of reliable models for the calibration of resting energy expenditure during inflammation [50,51].

## 5. Conclusions

Taken together, this study demonstrated that UCP1 production in AT is driven by IL6 and high RA activity. It occurred under conditions of insulin resistance and sought to counteract the high estimated metabolic and CV risk in those patients. Careful monitoring of metabolic health could be central for the prevention of CV events in non-diabetic patients with IL6-driven inflammation.

## Figures and Tables

**Figure 1 cells-10-01131-f001:**
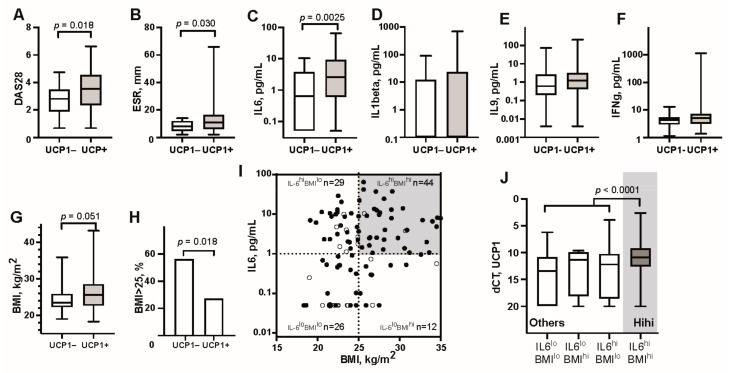
*UCP1* transcription in adipose tissue associates with high serum levels of IL-6 and overweight. *UCP1* transcription was analyzed by quantitative PCR in subcutaneous adipose tissue of 111 female patients with rheumatoid arthritis, among those 89 patients were UCP1 positive and 22 patients were UCP1-negative. (**A**) DAS28, (**B**) Erythrocyte sedimentation rate (ESR), Serum levels of (**C**) IL6, (**D**) IL1beta, (**E**) IL9, (**F**) IFNg, (**G**) Body mass index (BMI), (**H**) Frequency of overweight (BMI > 25 kg/m^2^). (**I**) Patient groups based on BMI and serum levels of IL6. BMI > 25 kg/m^2^ and IL6 > 1 pg/mL indicated high groups. Open circles indicate UCP1 negative cases. (**J**) The group combined high IL6 and BMI had significantly increased expression of UCP1. Boxplots show median, inter-quartile range and range. Statistical analysis was performed using non-parametric Mann–Whitney statistics and *p*-value < 0.05 are indicated.

**Figure 2 cells-10-01131-f002:**
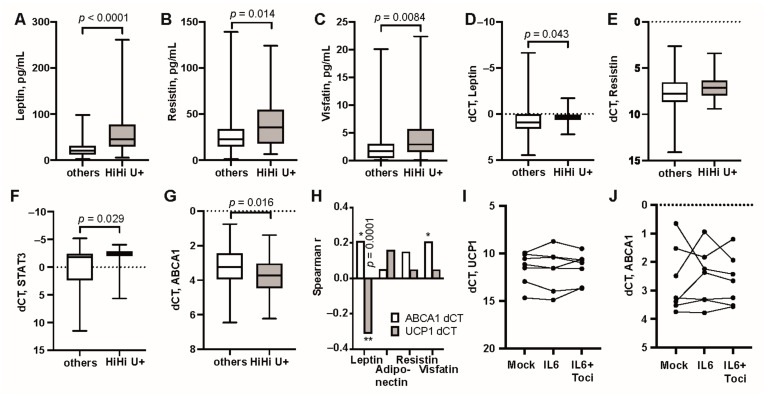
Activation of *UCP1* transcription down-stream of IL6 signaling. UCP1+ patients with high BMI and high IL6 (HiHiU+, *n* = 39) are compared to UCP1-negative patients (others, *n* = 69) by serum levels of leptin (**A**), resistin (**B**), visfatin (**C**); mRNA expression in adipose tissue of leptin (**D**), STAT3 (**E**), resistin (**F**), and ABCA1 (**G**). (**H**) Correlation between serum levels of adipokines and transcription of *ABCA1* (open bars) and *UCP1* (filled bars). mRNA expression of (**I**) *UCP1* and (**J**) *ABCA1* in adipose cells stimulated in vitro with IL6 +/− Tocilizumab. Boxplots show median, inter-quartile range and range. Statistical analysis was performed using the Mann–Whitney statistics. Correlations were performed with Spearman correlation and compared using Fisher r-to-z transformation (* *p* < 0.05, ** *p* < 0.01).

**Figure 3 cells-10-01131-f003:**
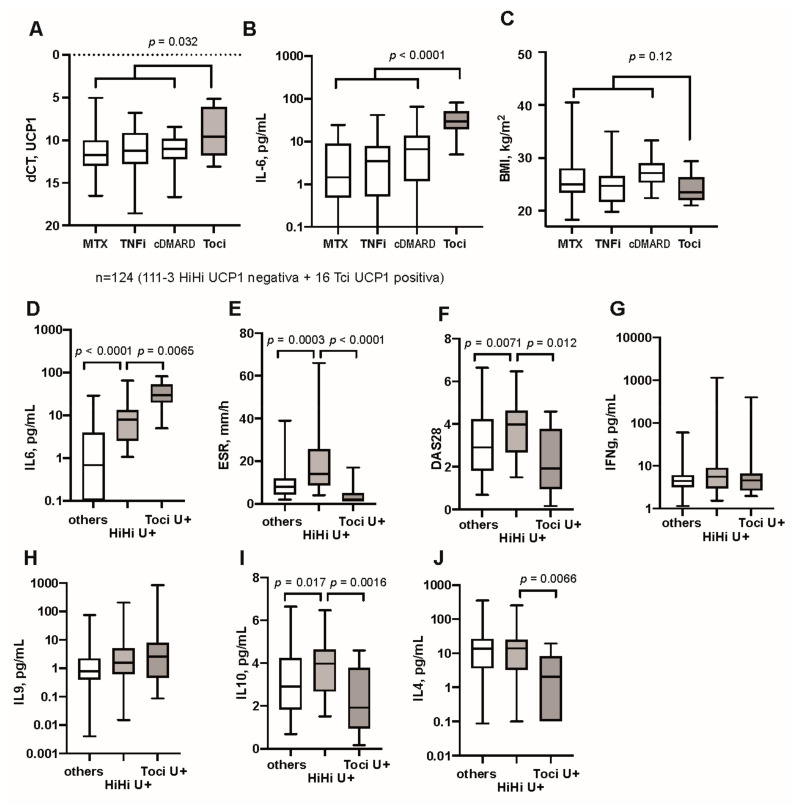
Inhibition of IL6 receptor signaling increases serum IL6 levels and *UCP1* transcription. Comparison of UCP1 positive patients with rheumatoid arthritis treated with methotrexate monotherapy (MTX, *n* = 45), TNFa inhibitors (TNFi, *n* = 28), conventional antirheumatic drugs (cDMARD, *n*=17), and IL6 receptor inhibitor tocilizumab (Toci, *n* = 18). (**A**) Transcription of *UCP1* in adipose tissue of (**B**) Serum levels of IL6. (**C**) Body mass index (BMI). Female RA patients were grouped by *UCP1* transcription into high UCP1 with high BMI and IL6 (HiHiU+, *n* = 39), tocilizumab-treated patients with high *UCP1* (TociU+, *n* = 16) and the remaining UCP1-negative patients (others, *n* = 69). (**D**) Serum levels of IL6, (**E**) erythrocyte sedimentation rate (ESR), (**F**) disease activity score (DAS28). Serum levels of (**G**) IFNg, (**H**) IL9, (**I**) IL10, and (**J**) IL4. Boxplots show median, inter-quartile range and range. Statistical analysis was performed using non-parametric Kruskal–Wallis statistics with post hoc analysis.

**Figure 4 cells-10-01131-f004:**
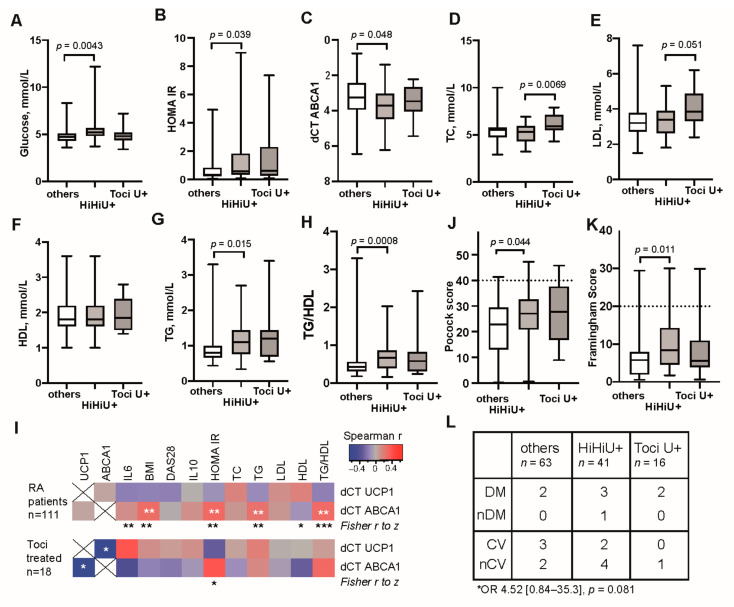
Unfavorable metabolic profile and increased cardiovascular risk despite high transcription of UCP1 female RA patients were grouped by UCP1 transcription into high UCP1 with high BMI and IL6 (HiHiU+, *n* = 39), tocilizumab-treated patients with high UCP1 (TociU+, *n* = 16) and the remaining UCP1-negative patients (others, *n* = 69). (**A**) Plasma glucose, (**B**) HOMA-IR, (**C**) total cholesterol (TC), (**D**) low-density lipids (LDL), (**E**) high-density lipids (HDL), (**F**) triglycerides (TG). (**G**) Triglycerides-to-HDL ratio. (**H**) mRNA expression of ABCA1. (**I**) Correlation of UCP1 and ABCA1 with inflammation and metabolic markers in RA patients (*n* = 111) and in tocilizumab-treated patients (*n* = 18). Analyses were done with Spearman’s correlation test and compared using Fisher r-to-z transformation (* *p* < 0.05, ** *p* < 0.01, *** *p* < 0.001). (**J**) Cardiovascular risk was estimated by Pocock’s score, (**K**) Framingham score. (**L**) Frequency of previous and diabetes mellitus (DM) and cardiovascular (CV) events in the groups and development of new events (nDM and nCV) at 5 years follow-up. Odds ratio (OR) was calculated between the groups’ HiHiU+ and others.

**Table 1 cells-10-01131-t001:** Patients with rheumatoid arthritis grouped by transcription of UCP1 in the adipose tissue.

	UCP1 Positive *n* = 89	UCP1 Negative *n* = 22	Tocilizumab *n* = 18
Age, years	56 (48–63)	61 (46–63)	65 (56–69)
Disease duration, years	7 (4–13)	8 (6–12)	16 (6–27)
Autoantibodies			
RF positive, *n* (%)	70 (80.4%)	15 (68.2%)	11 (69%)
ACPA positive, *n* (%)	56 (64.4%)	13 (59.1%)	9 (75%)
Tender joints, *n*	3 (0–10)	1 (0–5)	3 (1.5–8)
Swollen joints, *n*	2 (1–6)	1 (0–5.8)	2 (0–4.5)
VAS pain, mm	25 (12–61)	19 (9.5–33)	27 (21–41)
Methotrexate monotherapy, *n* (%)	33 (37%)	12 (54%)	
+ TNFa inhibitors, *n* (%)	23 (25%)	6 (27%)	
+ Other biologics, *n* (%)	12 (13%)	2 (9%)	
+ Conventional DMARDS, *n* (%)	16 (18%)	1 (5%)	
No DMARDS, *n* (%)	5 (6%)	1 (5%)	
Oral corticosteroids, *n* (%)	13 (15%)	1 (5%)	0
Levothyroxine, *n* (%)	10 (11%)	2 (9%)	1 (6%)
Estradiol, *n* (%)	3 (3%)	0	1 (6%)
Statins, *n* (%)	2 (2%)	0	3 (17%)
Diabetes type II, *n* (%)	4 (4%)	0	2 (11%)

Data are expressed as median and interquartile range. UCP1, uncoupling protein 1; RF, rheumatoid factor; ACPA, antibodies to citrullinated peptides; VAS, visual analogue scale; TNFa, tissue necrosis factor alpha; DMARDS, disease modifying antirheumatic drugs.

## Data Availability

The data presented in this study are available on request from the corresponding author. The data are not publicly available due to respect of privacy.

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
