# Peer review of "Impact of the Uncoupling Protein 1 on Cardiovascular Risk in Patients with Rheumatoid Arthritis"

_cells, 2021, doi:10.3390/cells10051131_

Round 1
Reviewer 1 Report
Manuscript ID: cells-1173054
Title: Impact of the Uncoupling Protein 1 on Cardiovascular Risk in Patients with Rheumatoid Arthritis
In this work, the authors aimed to investigate the clinical and metabolic connection between UCP1 (Uncoupling Protein 1) expression and cardiovascular health in female patients with rheumatoid arthritis (RA). Moreover, the effect of anti-rheumatic treatments on UCP1 expression was addressed.
The study indicates that high UCP1 transcription depends on IL6-driven process, associated with overweight and RA activity, and reflects primarily metabolic cardiovascular risk in female patients.
Besides the limitations of the study, highlighted in the discussion of the manuscript, the research design is appropriate to address the main aim and secondary objectives of the work. In addition, the results are clearly presented and support the main conclusions of the study.
For these reasons, I consider this manuscript suitable for publication in Cells journal, but the document can be improved according to the points mentioned below.
Major points:
- In the Discussion, the authors mention “In our study, we rely on measures of the transcription of UCP1 rather its protein levels and function”. The authors should explain/justify this sentence. Did the authors investigate the protein expression levels? Do the results follow the same tendency as the transcription measurements (mRNA)?
- The authors indicate that "IL6 could have a direct stimulatory effect on UCP1 transcription in adipocytes" (Results, lines 266-267). Nevertheless, the in vitro stimulation with IL6 did not show any change in the mRNA levels of UCP1. How do the authors explain the absence of a significant effect on UCP1? Could the short stimulation time (5h) be an explanation? Did the authors perform the assay at different time points?
Minor points:
- In the introduction, I recommend the authors define the abbreviation AT (adipose tissue), as this is the first time that appeared in the manuscript.
- In line 102 (Materials and Methods, Patients section), please rectify the abbreviation DMARD for disease modifying antirheumatic drugs (DMARDS), to uniformize the concept and abbreviation along the manuscript.
- In line 120 (Materials and Methods, Cardiovascular Risk Assessment section) please define the abbreviations BP (blood pressure) and TC (total cholesterol), as this is the first time mentioned in the manuscript.
- Please correct the typo in line 198 (Materials and Methods, Serological Parameters section). Insert a parenthesis rather than a comma: “and IFN(Sanquin, Amsterdam, the Netherlands)”.
- In line 205 (Materials and Methods, Serological Parameters section) please define the abbreviations LDL, HDL and TG. The definition only appears further in the legend of Figure 4.
- In line 210 (Materials and Methods, Statistical Analysis section), please define the abbreviation BMI. The definition only appears further in the legend of Figure 1.
- The legend of Figure 2 is incomplete. Please add D before the sentence “mRNA expression in adipose tissue of leptin”, and H before the sentence “Correlation between serum levels of adipokines and transcription of ABCA1 (open bars) and UCP1 (filled bars)”.
- In line 299 (Results, 3.2 section), please define the abbreviation TNFi (TNF inhibitors). The definition only appears further in the legend of Figure 3.
- In line 334 (Results, 3.3 section), please define the abbreviation DM (diabetes mellitus). The definition only appears further in the legend of Figure 4.
- The authors mention that “In the Pocock’s model, this difference did not reach significance (Kurskal-Wallis pos hoc, p=0.044). Nevertheless, according to the Material and Methods (Statistical Analysis, section), all the tests with p<0.05 were considered statistically significant.
- Please remove the parenthesis in reference 28 ((28), line 447, Discussion).
- Please correct the typo “improtance” (line 448).
Reviewer 2 Report
In this study, Lyngfelt L et al investigated the connection between UCP1 expression and cardiovascular health in patients with rheumatoid arthritis (RA) in a longitudinal observational study. They found that UCP1 transcription in adipose tissue associates with high serum levels of IL-6 and overweight and also revealed unfavourable Metabolic Profile in RA Patients with High Transcription of UCP1. While the topic of interest to the field, the design of the study needs improvement, and the data should be interpreted carefully, since majority are correlation study.
Major comments as following.
- The authors analyzed UCP1 expression as binary data, UCP1+ vs UCP-. From statistical standpoint, it would be more informative to further analyze UCP+ data as quantitative data.
- Abdominal wall adipose tissue biopsies were used in the study. However, UCP1 is mainly expressed in brown adipose tissue. The author should explain the scientific reason for this besides easy accessibility of tissue. How UCP1 in white adipose tissue correlated with whole-body thermogenic status?
- The author analyzed correlation between UCP1 in WAT and a number of serum cytokines (IL-6, IL-1beta, etc). While these cytokine in serum could be from any source in RA patient, it would be necessary to test expression of these cytokine in WAT and check their correlation with UCP1 levels.
- Fig 2I and J, the author should include positive control to show IL-6 stimulation does work well.
- Since in vitro experiment didn't support a causal connection between IL-6 and UCP1, the author are encouraged to word the conclusion carefully to avoid mislead readers. e.g. " The study shows that high transcription of UCP1 in the adipose tissue depends on IL6-driven processes..." in abstract.
Round 2
Reviewer 2 Report
The authors have addressed my concerns in revised manuscript.